# On the Pitfalls of Analyzing Individual Neurons in Language Models

**Omer Antverg**
Technion – Israel Institute of Technology
omer.antverg@cs.technion.ac.il

**Yonatan Belinkov**
Technion – Israel Institute of Technology
belinkov@technion.ac.il

## Abstract

While many studies have shown that linguistic information is encoded in hidden word representations, few have studied individual neurons, to show how and in which neurons it is encoded. Among these, the common approach is to use an external probe to rank neurons according to their relevance to some linguistic attribute, and to evaluate the obtained ranking using the same probe that produced it. We show two pitfalls in this methodology: 1. It confounds distinct factors: probe quality and ranking quality. We separate them and draw conclusions on each. 2. It focuses on encoded information, rather than information that is used by the model. We show that these are not the same. We compare two recent ranking methods and a simple one we introduce, and evaluate them with regard to both of these aspects.[1]

## 1 Introduction

Many studies attempt to interpret language models by predicting different linguistic properties from word representations, an approach called probing classifiers (Adi et al., 2017; Conneau et al., 2018, *inter alia*). A growing body of work focuses on individual neurons within the representation, attempting to show in which neurons some information is encoded, and whether it is localized (concentrated in a small set of neurons) or dispersed. Such knowledge may allow us to control the model's output (Bau et al., 2019), to reduce the number of parameters in the model (Voita et al., 2019; Sajjad et al., 2020), and to gain a general scientific knowledge of the model. The common methodology is to train a probe to predict some linguistic attribute from a representation, and to use it, in different ways, to rank the neurons of the representation according to their importance for the attribute in question. The same probe is then used to predict the attribute, but using only the $k$-highest ranked neurons from the obtained ranking, and the probe's accuracy in this scenario is considered as a measure of the ranking's quality (Dalvi et al., 2019; Torroba Hennigen et al., 2020; Durrani et al., 2020). We see this framework as exhibiting **Pitfall I**: Two distinct factors are conflated—the probe's classification quality and the quality of the ranking it produces. A good classifier may provide good results even if its ranking is bad, and an optimal ranking may cause an average classifier to provide better results than a good classifier that is given a bad ranking.

Another shortcoming of the current methodology, which we mark as **Pitfall II**, is the focus on encoded information, regardless of whether it is actually used by the model in its language modeling task. A few studies (Elazar et al., 2021; Feder et al., 2020) have considered this question, and shown that encoded information is not necessarily being used for language modeling, but these do not look at individual neurons. We argue that in order to evaluate a ranking, one should also examine if, and how, the $k$-highest ranked neurons are used by the model for the attribute in question, meaning that modifying them would change the model's prediction—but with respect to that attribute only. This would allow some control over the model's output, and grant us parameter-level explanations of the model's decisions.

In this work, we analyze three neuron ranking methods. Since the ranking space is too large (768! in BERT's case), these methods provide approximations to the problem and are non-optimal. Two of these methods—Linear (Dalvi et al., 2019) and Gaussian (Torroba Hennigen et al., 2020)—rely on an external probe to obtain a ranking: the first makes use of the internal weights of a linear probe, while the second considers the performance of a decomposable generative probe. The third is a

---

[1]Our code is available at: https://github.com/technion-cs-nlp/Individual-Neurons-Pitfalls

simple ranking method we propose, PROBELESS, which ranks neurons according to the difference in their values across labels, and thus can be derived directly from the data, with no probing involved.

We experiment with disentangling probe quality and ranking quality, by using a probe from one method with a ranking from another method, and comparing the different probe–ranking combinations. We expose the problematic nature of the current methodology (Pitfall I), by showing that in some cases, a suitable probe which is given an intentionally bad ranking, or a random one, provides higher accuracy than another which is given its allegedly optimal ranking. We find that while the GAUSSIAN method generally provides higher accuracy, its probe's selectivity (Hewitt & Liang, 2019) is lower, implying that it performs the probing task by memorizing, which improves probing quality but not necessarily ranking quality. We further find that GAUSSIAN provides the best ranking for small sets of neurons, while LINEAR provides a better ranking for large sets.

We then turn to analyzing which ranking selects neurons that are used by the model, by applying interventions on the representation: we modify subsets of neurons from each ranking and measure—using a novel metric we introduce—the effect on language modeling w.r.t to the property in question. We highlight the need to focus on used information (Pitfall II): even though PROBELESS does not excel in the probing scenario, it selects neurons that are used by the model, more so than the two probing-based rankings. We find that there is an overlap between encoded information and used information, but they are not the same, and argue that more attention should be given to the latter.

We primarily experiment with the M-BERT model (Devlin et al., 2019) on 9 languages and 13 morphological attributes, from the Universal Dependencies dataset (Zeman et al., 2020). We also experiment with XLM-R (Conneau et al., 2020), and find that most of our results are similar between the models, with a few differences which we discuss. Our experiments reveal the following insights:

- We show the need to separate between probing quality and ranking quality, via cases where intentionally poor rankings provide better accuracy than good rankings, due to probing weaknesses.

- We present a new ranking method that is free of any probes, and tends to prefer neurons that are being used by the model, more so than existing probing-based rankings.

- We show that there is an overlap between encoded information and used information, but they are not the same.

## 2 NEURON RANKINGS AND DATA

We begin by introducing some notation. Denote the word representation space as $H \subseteq \mathbb{R}^d$ and an auxiliary task as a function $F : H \to Z$, for some task labels $Z$ (e.g., part-of-speech labels). Given a word representation $h \in H$ and some subset of neurons $S \subseteq \{1, ..., d\}$, we use $h_S$ to denote the subvector of $h$ in dimensions $S$. For some auxiliary task $F$ and $k \in \mathbb{N}$, we search for an optimal subset $S^*$ such that $|S^*| = k$ and $h_{S^*}$ contains more information regarding $F$ than any other subvector $h_{S'}, |S'| = k$. For the search task, we define *neuron-ranking* as a permutation $\Pi(d)$ on $\{1, ..., d\}$ and consider the subset $\Pi(d)_{[k]} = \{\Pi(d)_1, ..., \Pi(d)_k\}$. One may wish to find an optimal ranking $\Pi^*(d)$ such that $\forall k, \Pi^*(d)_{[k]}$ is the optimal subset with respect to $F$. However, finding an optimal ranking, or even an optimal subset, is NP-hard (Binshtok et al., 2007). Thus, we focus on several methods to produce rankings, which provide approximations to the problem, and compare them.

### 2.1 RANKINGS

The ranking methods we compare include two rankings obtained from prior probing-based neuron-ranking methods, and a novel ranking we propose, based on data statistics rather than probing.

**LINEAR** The first method, henceforth LINEAR (named linguistic correlation analysis in Dalvi et al. 2019),[2] trains a linear classifier on the representations to learn the task $F$. Then, it uses the trained classifier's weights to rank the neurons according to their importance for $F$. Intuitively, neurons with a higher magnitude of absolute weights should be more important, or contain more relevant information, for solving the task. Dalvi et al. (2019) showed that their method identifies important neurons through probing and ablation studies, and found that while the information is distributed across neurons, the distribution is not uniform, meaning it is skewed towards the top-ranked neurons. In this work, we use a slightly modified version of the suggested approach (Appendix A.1).

---

[2] A small enhancement to the algorithm was presented in Durrani et al. (2020).

**GAUSSIAN** The second method, henceforth GAUSSIAN (Torroba Hennigen et al., 2020), trains a generative classifier on the task $F$, based on the assumption that each dimension in $\{1, ..., d\}$ is Gaussian-distributed. Then, it makes use of the decomposability of the multivariate Gaussian distribution to greedily select the most informative neuron, according to the classifier's performance, at every iteration. This way we obtain a full neuron ranking after training only once, while applying this greedy method to LINEAR would require retraining the probe $d!$ times, which is clearly infeasible. Torroba Hennigen et al. (2020) found that most of the tasks can be solved using a low number of neurons, but also noted that their classifier is limited due to the Gaussian distribution assumption.

**PROBELESS** The third neuron-ranking method we experiment with is based purely on the representations, with no probing involved, making it free of probing limitations (Belinkov, 2021) that might affect ranking quality. For every attribute label $z \in Z$, we calculate $q(z)$, the mean vector of all representations of words that possess the attribute and the value $z$. Then, we calculate the element-wise difference between the mean vectors,

$$r = \sum_{z,z' \in Z} |q(z) - q(z')|, \qquad r \in \mathbb{R}^d \qquad (1)$$

and obtain a ranking by arg-sorting $r$, i.e., the first neuron in the ranking corresponds to the highest value in $r$. For binary-labeled attributes, this is simply the difference in means. In the general case, PROBELESS assigns high values to neurons that are most sensitive to a given attribute. We note that PROBELESS is very fast to use, as we are only limited by averaging and sorting, as opposed to training a classifier in LINEAR or the expensive greedy algorithm of GAUSSIAN.

## 2.2 DATA AND MODELS

Throughout our work, we follow the experimental setting of Torroba Hennigen et al. (2020): we map the UD treebanks (Zeman et al., 2020) to the UniMorph schema (Kirov et al., 2018) using the mapping by McCarthy et al. (2018). We select a subset of the languages used by Torroba Hennigen et al. (2020): Arabic, Bulgarian, English, Finnish, French, Hindi, Russian, Spanish and Turkish, to keep linguistic diversity. The tasks we experiment with are predictions of morphological attributes from these languages. Full data details are provided in Torroba Hennigen et al. (2020) and further data preparation steps are detailed in Appendix A.2. We process each sentence in pre-trained M-BERT and XLM-R (unless stated otherwise, all results are with M-BERT), and take word representations from layers 2, 7 and 12 of each model, to see if there are different patterns in the beginning, middle and end of the models. We end up with a total of 156 different configs (language $\times$ attribute $\times$ layer) to test for each model. For words that are split during tokenization, we define their final representation to be the average over their sub-token representations. Thus, each word has one representation for each layer, of dimension $d = 768$. We do not mask any words throughout our work.

## 2.3 OVERLAPS

Before evaluating our rankings in different scenarios, we first characterize them by looking at the 100-highest ranked neurons (out of 768) from different rankings, across different configs.

**Some neurons are important for an attribute across languages** Since we work with multilingual models, we expect to see overlap in the selected neurons for one attribute across different languages. Fig. 1 shows that for PROBELESS this is indeed the case, as some attributes share a large number of important neurons across languages. For example, number in Spanish and number in French share 70 of their 100 most important neurons, where the expected number for overlap of two random selections of neurons is only 13 (Appendix A.3). Compared to the other two rankings (Appendix A.4), PROBELESS is the most consistent across languages, while GAUSSIAN rarely shows consistency, which may be a weakness.

**Some neurons are unanimously important** By looking at the overlaps between important neurons selected by different rankings for the same config, we observe that for all configs, the overlap between all three rankings surpasses the expected number (which is $\sim 1.69$; Appendix A.3), meaning there are neurons that are recognized by all three rankings as important.

We further see that in most cases, the greatest overlap is between LINEAR and PROBELESS. We find it reasonable, as both of them aim to select neurons that separate classes the best—one by a classifier

and the other by data statistics—while GAUSSIAN takes a different approach, assuming a Gaussian distribution and selecting neurons only by performance. Examples from 8 configs are shown in Fig. 2.

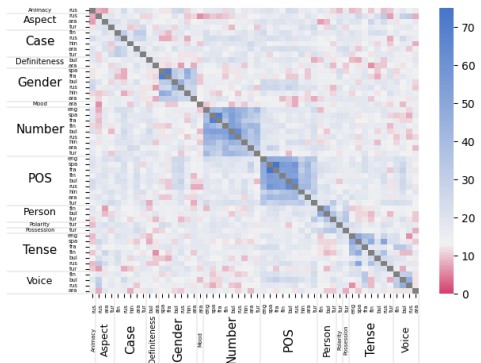

Figure 1: Layer 7 neurons overlap, using PROBELESS ranking. Blue squares are above the expected overlap between 2 rankings (Appendix A.3), red are below. Major ticks are attributes, minor are languages.

Figure 2: Layer 2 neurons overlap between every pair of rankings, from 8 randomly selected configs. Gray and black dashed lines show the expected overlap between 2 and 3 random rankings, respectively (Appendix A.3).

**There are more overlaps in XLM-R** Performing the same analysis across languages on XLM-R (Appendix A.4), the overlap size is at least as the expected one between all config pairs, and is usually greater. It may imply that XLM-R's representations can be pruned more easily than M-BERT's, since some neurons encode multiple attributes, and there is greater redundancy among the others.

## 3 PITFALL I: CLASSIFIERS VS. RANKINGS

We now turn to evaluating the rankings, and present the pitfalls in doing so. Given some ranking $\Pi(d)$, we would like to evaluate how well it sorts the neurons for the task $F$. Our first ranking-evaluation approach is the standard probing approach from previous work (Dalvi et al., 2019; Torroba Hennigen et al., 2020), where we expose the classifier to a subvector of the representation and evaluate how well it predicts the task. However, while previous work conflated rankings and classifiers—Pitfall I—we are more careful: we separate the two, and pair each ranking with two classifiers, meaning that at least one of them is completely unrelated to the ranking.

Formally, for an increasing $k \in \mathbb{N}$, we train a classifier $f : H_k \to Z$ to predict the task label, $F(h)$, solely from $h_{\Pi(d)_{[k]}}$ (the subvector of the representation $h$ in the top $k$ neurons in ranking $\Pi$), ignoring the rest of the neurons.[3] The assumption is that the better $f$ performs, the more task-relevant information is encoded in $h_{\Pi(d)_{[k]}}$. Yet, the behaviour of $f$ itself might affect results and conclusions about the ranking. Thus, we take classifier capabilities into consideration when analyzing results, and also measure selectivity (§3.1.1). The process is further illustrated in Appendix A.5.

### 3.1 EXPERIMENTAL SETUP

As classifiers, we experiment with both classifiers used by the first two ranking methods (LINEAR and GAUSSIAN). We use the hyperparameters reported in Durrani et al. (2020) and Torroba Hennigen et al. (2020) for training the classifiers. As rankings, we experiment with the 3 ranking methods described in §2.1. For each, we use the original ranking it produces and its reversed version, referred to as top-to-bottom and bottom-to-top, respectively. To those we add a random ranking baseline, resulting in 7 different rankings overall. We compare all classifier–ranking combinations for each $k$.

Since both of the first two ranking methods are inherently tied to the classifier that was used to generate them, and the third ranking is a classifier-neutral ranking, it can be used for a fair comparison between the classifiers.

---

[3]We only probe into representations of words that possess the attribute, e.g., if the attribute is gender we do not probe into the representation of the word "pizza".

### 3.1.1 METRICS

**Accuracy**   First, we measure the accuracy of the probe's predictions. Since we experiment with many different configs, we use the Wilcoxon signed-rank test (Wilcoxon, 1992) as a statistical significance test to determine whether a certain combination of a classifier and a ranking is statistically significantly better than another combination.

**Selectivity**   We also evaluate our probes by selectivity (Hewitt & Liang, 2019), defined as the difference between the classifier's accuracy on the actual probing task and its accuracy on predicting random labels assigned to word types, called a control task. Low selectivity implies that the probe can memorize the word-type–label pair, and so high accuracy in the probing task does not necessarily entail the presence of the linguistic attribute. Thus, we prefer probes that are both accurate and selective.

### 3.2 RESULTS

Across the 156 configs we experiment with, we observe three different accuracy patterns, demonstrated in Figs. 3a-3c. In these figures, each color represents a combination of a classifier and a ranking, where a solid line is used for the top-to-bottom version of the ranking and a dotted line is for the bottom-to-top version of it, and a dashed line is used for the random ranking. Almost half of the configs follow the Standard pattern (Fig. 3a), in which all top-to-bottom rankings are always better than the random ranking, which is always better than all bottom-to-top rankings. The other half consists of two surprising patterns, that demonstrate the inherent flaws in this ranking-evaluation approach. In the G>L pattern (Fig. 3b), the GAUSSIAN classifier performs exceptionally well, providing higher accuracy (after a certain point) using a random or even a bottom-to-top ranking, than the LINEAR classifier using its top-to-bottom ranking. In the L>G pattern (Fig. 3c), the GAUSSIAN classifier fails quickly, and thus the LINEAR classifier provides higher accuracy using a random or bottom-to-top ranking than the GAUSSIAN classifier using its top-to-bottom ranking.

Fig. 3d shows a t-SNE (van der Maaten & Hinton, 2008) projection after performing K-means clustering on our 156 accuracy results, where each point represents accuracy results from one config (details on clustering procedure are in Appendix A.6). It shows three clusters of configs, that correspond to the three distinct patterns. On XLM-R we see very similar results, and most configs follow the same pattern in each model (Appendix A.8). We now turn to analyze these results.

### 3.2.1 RANKING METHODS ARE INHERENTLY CONSISTENT

In most configs, each classifier provides better accuracy using a top-to-bottom ranking (solid lines) compared to the bottom-to-top version of the same ranking (same color, dotted line), and the random ranking (dashed lines) is in between. This is also seen in our statistical significance tests (Appendix A.7). We conclude that even if they are not optimal, all ranking methods we consider generally rank task-informative neurons higher than non-informative ones.

### 3.2.2 WHICH CLASSIFIER IS BETTER?

In the Standard and G>L patterns (Figs. 3a, 3b), GAUSSIAN achieves better accuracy than LINEAR when both of them use the same ranking (including top-to-bottom LINEAR), especially when using small sets of neurons. Our statistical significance tests (Appendix A.7) show that GAUSSIAN performs significantly better than LINEAR with 6 out of the 7 rankings we tried when using 10 neurons, with 5 rankings when using 50 neurons, and with 5 rankings when using 150 neurons. We now turn to analyze what makes GAUSSIAN more successful, and show some exceptions.

**GAUSSIAN is memorizing**   Across all configs, LINEAR provides higher selectivity than GAUSSIAN using any ranking, after a certain point (Appendix A.8). This means that GAUSSIAN tends to memorize the word-type–label pair when solving the task. While this is apparent in all configs, we note that specifically in the part-of-speech attribute there is a large portion of function words, i.e., closed set labels (e.g., pronouns, determiners), meaning that memorization can significantly help solve the task. Thus, most configs involving part of speech belong to the G>L pattern. Since memorization is a trait of the classifier and not of the ranking, this pattern demonstrates the problematic nature of the current ranking-evaluation approach, as the results are highly dependent on the probe.

**LINEAR is more stable**   On the other hand, pattern L>G shows that there are certain configs where GAUSSIAN is struggling to model the distribution, resulting in mediocre accuracy results—which

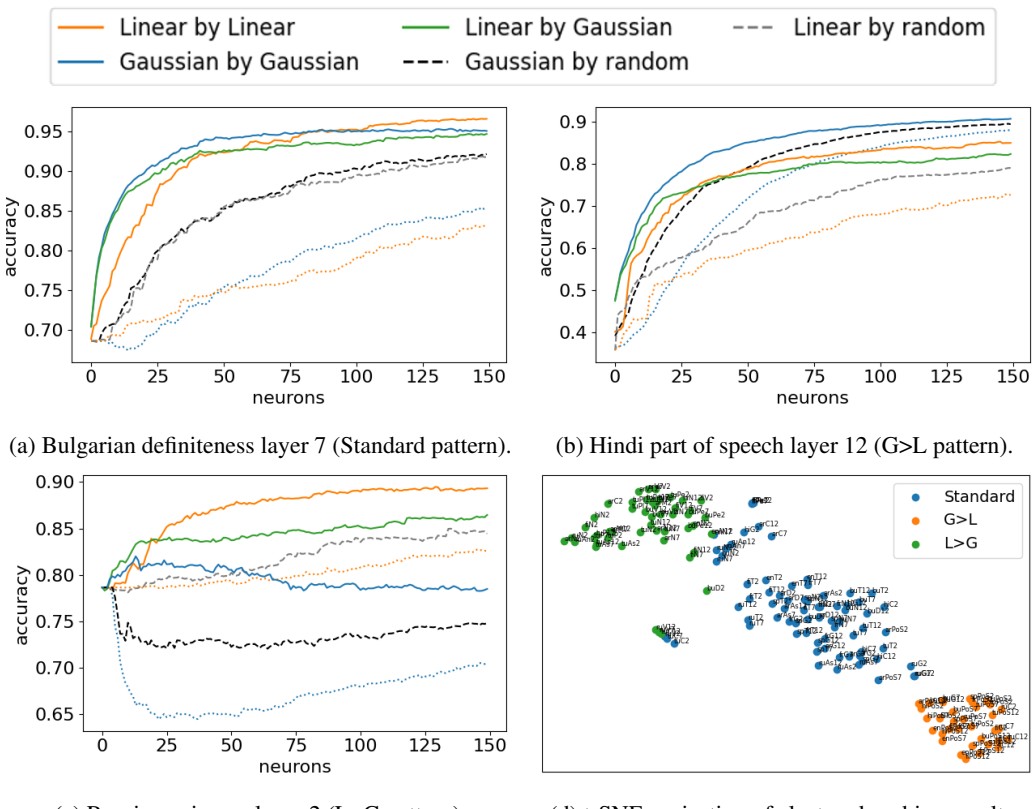

(a) Bulgarian definiteness layer 7 (Standard pattern).

(b) Hindi part of speech layer 12 (G>L pattern).

(c) Russian animacy layer 2 (L>G pattern).

(d) t-SNE projection of clustered probing results.

Figure 3: Clustering of the three different patterns (3d), and an example of each of the patterns (3a–3c). Solid lines are top-to-bottom rankings; dashed are random rankings; dotted are bottom-to-top rankings. "X by Y" means classifier X using ranking Y. Some lines are omitted for clarity; complementing figures can be found in Appendix A.8.

even start decreasing at some point—sometimes even below majority baseline, as seen in Fig. 3c. This has also been mentioned in Torroba Hennigen et al. (2020), where it was shown that in those configs, there are only a few (or no) dimensions that are informative for the attribute and are Gaussian-distributed. Thus, the GAUSSIAN classifier tries to model these distributions with the wrong tools, and fails. Poor modeling then leads to wrong predictions and low accuracy. In general, LINEAR behaves similarly across configs, making it more stable.

### 3.2.3 WHICH RANKING IS BETTER?

When looking at rankings, we would like to compare performance of the same classifier, using different rankings. We would expect that each classifier would perform best when using the ranking it has generated. However, this is not always the case. As we can see in all patterns in Fig. 3, for small sets of neurons, LINEAR actually achieves better accuracy when using GAUSSIAN's ranking (solid green) than its own ranking (solid orange). As the number of neurons increases, at some point its accuracy with its own ranking becomes higher than with GAUSSIAN's ranking.

We suggest two explanations for this phenomenon: First, due to its greediness, the GAUSSIAN ranking is not guaranteed to provide the optimal subset. For a subset of size 1, it goes over all possibilities, but as the size grows there are more subsets that are not taken into consideration in the algorithm, so it is more likely to miss the best sets. Second, GAUSSIAN assumes the embedding distribution to be Gaussian. On dimensions which are not Gaussian-distributed, it makes a less accurate evaluation of the contribution of each neuron. So, if a neuron is informative towards the attribute but is not Gaussian-distributed, its addition to the selected neurons set is unlikely to improve performance, and thus it is not selected. This is a problem with a performance-based selection criterion, where the selection of neurons depends on the performance of the probe.

To summarize, it seems that GAUSSIAN is good at selecting specific informative neurons, but misses the rest. While LINEAR's ranking is not optimal (it is definitely worse then GAUSSIAN's on small sets), it does seem to be more stable on different sizes. PROBELESS provides decent performance (and is inherently consistent), but is usually behind the other two.

## 4   PITFALL II: ENCODED INFORMATION VS. USED INFORMATION

The variance of results in our probing experiments can mostly be attributed to probing limitations (Hewitt & Liang, 2019; Belinkov, 2021), and emphasizes the need to distinguish between two properties: the probe's classification quality, and the neuron-ranking quality. To isolate the latter, and to shed light on which ranking prefers neurons that are actually used by the model for the attribute in question (which is ignored by previous work—Pitfall II), we take a second ranking-evaluation approach: we intervene by modifying the representation in the neurons selected by the ranking, and observe if, and how, our intervention affects the language model output. This approach is more of a causal one, inspired by similar prior work (Giulianelli et al., 2018; Elazar et al., 2021; Feder et al., 2020; Lovering et al., 2021; Ravfogel et al., 2021). We note that in this section, we use only the ranking itself, detaching it from any probes, thus removing classification quality from ranking comparisons.

Formally, for a representation $h \in H$, ranking $\Pi(d)$ (corresponding to an attribute $F$) and an increasing $k \in \mathbb{N}$, we intervene by modifying $h$ only in the $\Pi(d)_{[k]}$ neurons, and observe the effect our intervention had on the model's output—the word prediction (given the modified representation).[4] For vocabulary $\mathcal{V}$, we divide the model to two components: $E : \mathcal{V} \to H$ and $D : H \to \mathcal{V}$, such that for interventions in layer $i$, $E$ is composed of all of the layers of the model up to (including) $i$, and $D$ is composed of all of the rest of the layers, including the classification head. After receiving a representation $h = E(w)$ for word $w \in \mathcal{V}$, we modify $h$ to get a new representation $h'$. If $D(h) \neq D(h')$, then $D$ is using the modified information. The process is illustrated in Appendix A.9.

However, knowing that the information is being used is not enough; we would like to know to what purpose it is being used, and to verify that it only affects the specific attribute we are interested in. Thus, we perform a finer-grained analysis, and check if $D(h')$ is similar, to some extent, to $D(h)$. For that, we define a lemmatizer $L : \mathcal{V} \to \mathcal{V}$, which maps words to their lemmas, and an analyzer $A : \mathcal{V} \to Z$, which maps words to their task labels. Our goal is to intervene such that $L(D(h)) = L(D(h'))$, but $A(D(h)) \neq A(D(h'))$. For example, if we intervene for tense, we would like the word "sleeps" to become "slept". If this is the case, it implies that we have successfully identified where the task-relevant information that $D$ uses is encoded, and how it is being used.

### 4.1   INTERVENTION METHODS

We consider two methods for modifying $h_{\pi(d)_k}$, and compare them.

**Ablation**   A common modification method is trying to remove the information by ablating some neurons (Morcos et al., 2018; Bau et al., 2019; Lakretz et al., 2019), meaning we set $h_{\pi(d)_k} = 0$. By that we aim to erase the information encoded in $h_{\pi(d)_k}$.

**Translation**   For a word $w \in \mathcal{V}$ with attribute label $z \in Z$, we attempt to translate its representation (in the geometric sense) to produce a word with attribute label $z' \in Z, z \neq z'$ by taking a step in the direction of $z'$, where bigger steps are applied to neurons that are marked as more important for the attribute. Formally, we apply the following protocol:

1. We calculate $q(z)$ and $q(z')$ as in eq. (1).
2. We set
$$h_{\Pi(d)_{[k]}} = h_{\Pi(d)_{[k]}} + \alpha_k(q(z')_{\Pi(d)_{[k]}} - q(z)_{\Pi(d)_{[k]}}) \tag{2}$$
   where $\alpha \in \mathbb{R}^d$ is a log-scaled coefficients vector in the range $[0, \beta]$, such that the coefficient of the highest-ranked neuron is $\beta$ and that of the lowest-ranked neuron is 0, and $\beta$ is a hyperparameter.

Note that the rest of the neurons—those not in $\Pi(d)_{[k]}$—remain unaffected. Using this protocol, we give each neuron its own special treatment—an approach that was not applied before (as far we know). This can be seen as a generalization of Gonen et al. (2020).

---

[4]We apply the intervention on representations of all words that possess the attribute in the sentence.

## 4.2 Experimental setup

We handle the data the same way as in our probing experiments. However, since we analyze the model's predictions—which may be different from the original input—we do not have gold morphology labels anymore. Thus, for morphologically analyzing the model's predictions ($L$ and $A$), we use spaCy (Honnibal et al., 2020). Out of the languages we used in our probing experiments, in this section we use only those that are supported by spaCy (English, Spanish and French). We calculate $q(z)$ based on the entire training set, and perform our interventions on the test set. We compare the same 7 rankings we used in our probing experiments (§ 3.1).

## 4.3 Metrics

**Error rate** For our intervention experiments, we first measure the error rate of the language model. We want error rate to be high, since high error rate means we modified parts of the representation that have been used by the model in its prediction.

**Correct Lemma, Wrong Value (CLWV)** While inspecting predictions that are wrong after intervening ($D(h') \neq w$, where $w$ is the true word), we categorize them by $L(D(h'))$ and $A(D(h'))$. If our intervention were successful, meaning we changed only the word's specific attribute, and not other information, then we expect to see $L(D(h)) = L(D(h'))$ and $A(D(h)) \neq A(D(h'))$; that is, correct lemma but wrong value (CLWV). For example, if the word "makes" becomes "made" when intervening for tense, then it is considered as a correct type of error, but if it becomes "make" or "prepared" it does not. Thus, we define CLWV as the portion of those errors out of all predictions.

## 4.4 Results

### 4.4.1 Ablation is not effective

Across most configs, about 400 neurons from layer 2 and 200–300 neurons from layers 7 and 12 can be ablated without any implications on the output, meaning error rate remains the same; an example is shown in Appendix A.10. Moreover, when error rate does grow, CLWV is very low. By qualitatively analyzing those errors we saw that most predicted words are common words, e.g., "and", "if" in English. After ablating 600–700 (80%–90%) neurons from the representations, we observe a lot of errors, but most of them are because the word is predicted as nonsensical punctuation. Another major concern is that in some configs, ablating by a bottom-to-top ranking provides better results than by the top-to-bottom version of the same ranking. In general, there are no distinct differences between the rankings. Thus, from here on we focus on translation rather than ablation.

### 4.4.2 Translation is effective

Across all translation experiments (Fig. 4 shows one example, more are in Appendix A.11), CLWV increases until a certain saturation point, after which it remains constant or drops a little.[5] This means that we reached neurons that are not relevant for the attribute, and modifying them can result in loss of other information—error rate grows while CLWV does not. Thus, we are interested in the CLWV value at the saturation point (higher is better), and in the number of neurons modified at the saturation point (lower is better). We would also like the difference between the error rate and CLWV at the saturation point to be as small as possible. All terms considered, we perform a sweep search on the values of $\beta$ in the range $[1, 12]$ on a dev set. We find that low $\beta$ values provide low CLWV, while high values provide higher CLWV but also widen the gap between error rate and CLWV. We find $\beta = 8$ to be a balanced point, and thus report test results with $\beta = 8$ in three configs in Table 1, and the rest of the configs in Appendix A.11. The results for XLM-R are given in Appendix A.12.

Compared to ablation, translating a relatively small number of neurons results in a higher error rate, and these errors are closer to what we would expect. For example, translating only 50 neurons selected by PROBELESS in Spanish gender layer 2 results in 37% CLWV and 49% error rate, while ablating 50 neurons from the same config and ranking gives 0% CLWV errors and only 1% of error rate. We further note that unlike in ablation experiments, here our rankings are inherently consistent: across all configs, all top-to-bottom rankings perform better than the rest, while random rankings'

---

[5]We define "saturation point" as the first point from which there are two consecutive points where the value increase is by a factor lower than 1.05.

error rate sometimes increases a little, and bottom-to-top rankings do not manage to affect the model's output at all (Fig. 4 is one example, more are in Appendix A.11).

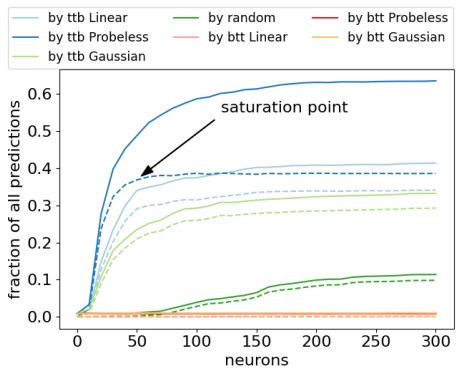

Figure 4: Spanish gender layer 2, translation results with $\beta = 8$. Solid lines are error rates, dashed are CLWVs. ttb and btt stand for top-to-bottom and bottom-to-top, respectively.

Table 1: CLWV value at saturation point and number of neurons modified at the saturation point, using the translation method, with $\beta = 8$. In each cell, the three lines refer to layers 2, 7 and 12 respectively.

|  | LINEAR | GAUSSIAN | PROBELESS |
|---|---|---|---|
| English tense | 0.39, 60 
 0.37, 50 
 0.51, 60 | 0.26, 150 
 0.34, 70 
 0.41, 120 | 0.38, 30 
 0.34, 30 
 0.46, 30 |
| Spanish number | 0.28, 110 
 0.26, 50 
 0.23, 150 | 0.19, 100 
 0.20, 40 
 0.16, 140 | 0.35, 60 
 0.25, 30 
 0.40, 80 |
| Spanish gender | 0.29, 50 
 0.29, 50 
 0.26, 130 | 0.25, 80 
 0.31, 50 
 0.16, 110 | 0.37, 50 
 0.33, 30 
 0.35, 60 |

### 4.4.3 PROBELESS IS THE MOST EFFECTIVE RANKING FOR INTERVENTIONS

A clear trend from our M-BERT results (Table 1, Fig. 4 and Appendix A.11) is that in most cases, PROBELESS achieves higher CLWV values, and does so using a smaller number of neurons, than the other two rankings. Furthermore, its error rate is significantly higher than the other two. This implies that PROBELESS tends to select neurons that are being used by the model, more so than the other rankings. However, while it does select neurons that are relevant for the attribute in question (CLWV is relatively high), it also tends to select neurons that are used by the model for other kinds of attributes (the difference between error rate and CLWV is relatively high). Among LINEAR and GAUSSIAN, LINEAR seems to have the upper hand, with higher CLWV values in most configs. This provides another evidence that the superiority of GAUSSIAN in the probing experiments may be due to the quality of its classifier, and specifically its memorization ability, rather than the quality of the ranking it produces, as here only the ranking affects results. In XLM-R, it seems that LINEAR and PROBELESS both have the lead, with GAUSSIAN falling behind (Appendix A.12). We perform additional experiments on monolingual models (Appendix A.13), where PROBELESS is again superior.

## 5 DISCUSSION AND CONCLUSION

In this work, we show two pitfalls with the common approach for ranking neurons according to their importance for a morphological attribute, and compare different ranking methods that follow this approach. We show that to evaluate a ranking in a probing scenario, one should separate between the ranking itself and the quality of the classifier that is using the ranking—Pitfall I. While previous work concentrated on encoded information—Pitfall II—we show that it is not the same as information used by a model, by showing that GAUSSIAN is inferior in the interventions scenario, in contrast to our probing results. This implies that high probing accuracy does not necessarily entail that the information is actually important for the model. This conclusion is also present in prior work (Elazar et al., 2021; Feder et al., 2020; Ravfogel et al., 2021), but it has been largely neglected in studies of individual neurons via probes. We propose a new, fast-to-use ranking method that relies solely on the data, without training any auxiliary classifier, and show that it is valid, and prefers neurons that are being used by the model, more so than other ranking methods. We also propose a method for intervening within the model's representations such that it transforms the output in a desired way.

In our intervention experiments, modifying too many neurons results in more errors that are not related to the true word. This proves the importance of looking into individual neurons, especially when trying to intervene in the inner workings of the model. For example, Gonen et al. (2020) try to change the language of a word by intervening with the representation, using the same translation method we use, but with the same coefficient for every neuron, and on the entire representation. Our results imply that they may get better results by using our finer-grained method.

**Ethics statement**   Our work contributes to the effort of improving the interpretability of language models, and more generally of neural networks. Better explanations and controls over a model's outputs can ameliorate its fairness, for example in the case of gender bias: our intervention method can guide users on how to reduce such biases, by pointing to model components (neurons) responsible for gender and offering intervention methods to control model behavior w.r.t a particular property. On the other hand, malicious actors could use such capability to increase discrimination. Exposing the capabilities may also help develop defense mechanisms.

**Reproducibility statement**   All our results are reproducible using the code repository we will release. All experimental details, including hyperparameters, are reported in §3.1, 4.1 and 4.2. As language models, we used the implementation of the transformers library (Wolf et al., 2020). We performed our experiments on NVIDIA RTX 2080 Ti GPU. All data preparation details are reported in §2.2 and Appendix A.2.

ACKNOWLEDGMENTS

We thank Lucas Torroba Hennigen for his helpful comments. This research was supported by the ISRAEL SCIENCE FOUNDATION (grant No. 448/20) and by an Azrieli Foundation Early Career Faculty Fellowship. YB is supported by the Viterbi Fellowship in the Center for Computer Engineering at the Technion.

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

# A APPENDIX

## A.1 LINEAR METHOD MODIFICATION

In Dalvi et al. (2019), after the probe has been trained, its weights are fed into a neuron ranking algorithm. However, we observed that the original algorithm distributes the neurons equally among labels, meaning that each label would contribute the same number of neurons at each portion of the ranking, regardless of the amount of neurons that are actually important for this label. For example, if for label A there are 10 important neurons and for label B there are only 2, then the first 10 neurons in the ranking would consist of 5 neurons for A and 5 for B, meaning that 3 non-important neurons are ranked higher than 5 important ones. Thus, we chose a different way to obtain the ranking: for each neuron, we compute the mean absolute value of the $|Z|$ weights associated with it, and sort the neurons by this value, from highest to lowest. In early experiments we found that this method empirically provides better results, and is more adapted to large label sets.

## A.2 DATA PREPARATION

We remove any sentences that would have a sub-token length greater than 512, the maximum allowed for M-BERT, the language model we use for generating representations. As in Torroba Hennigen et al. (2020), we remove attribute labels that are associated with fewer than 100 word types in any of the data splits. This mostly removes function words, and we found it makes it harder for probes to use memorization for solving the task. The morphological attributes we experiment with include (in UniMorph annotations): Animacy, Aspect, Case, Definiteness, Gender and Noun Class, Mood, Number, Part of Speech, Person, Polarity, Possession, Tense and Voice.

## A.3 EXPECTED OVERLAP BETWEEN RANDOM RANKINGS

Given $i$ rankings, we calculate the expected size of overlap between the first $M$ neurons across all rankings:

For selecting $M$ neurons from the range $\{1, ..., N\}$, let $C_i \in \mathbb{R}^{N \times N \times N}$ be a matrix such that in $C_i[n, m, k]$ we keep the number of possibilities to select $m$ neurons from the range $[n]$ such that exactly $k$ different neurons are selected by all $i$ rankings, where $k \leq m \leq n$ and $n, m, k > 0$. For calculating $C_i[n, m, k]$ we first select the $k$ neurons from range $[n]$ that are selected by all $i$ rankings, thus $\binom{n}{k}$ possibilities. Then, for selecting the rest of the neurons, each ranking has to select $m - k$ neurons from the remaining $n - k$ neurons, so there are $\binom{n-k}{m-k}^i$ possibilities. From these, we want to substract the number of possibilities in which there is at least one neuron that is selected by all rankings, which is $\sum_{j=1}^{m-k} C_i[n - k, m - k, j]$. Concluding, we compute $C_i[n, m, k]$ by:

$$C_i[n, m, k] = \binom{n}{k} \left( \binom{n-k}{m-k}^i - \sum_{j=1}^{m-k} C_i[n - k, m - k, j] \right) \tag{3}$$

after initializing $C[1, 1, 1] = 1$. Then, we calculate the expected number of overlapping neurons by:

$$E_i(n, m) = \frac{\sum_{k=1}^{m} k \times C[n, m, k]}{\binom{n}{m}^i} \tag{4}$$

since $\frac{C[n,m,k]}{\binom{n}{m}^i}$ is the probability to have exactly $k$ overlapping neurons. We thus get $E_2(768, 100) \approx 13.02$ and $E_3(768, 100) \approx 1.69$.

## A.4 OVERLAPS

Fig. 5 shows overlaps between the 100 most important neurons chosen by LINEAR and GAUSSIAN for different configs. Both of them provide less overlaps then PROBELESS, with GAUSSIAN having almost no overlaps at all, showing its inconsistency across languages.

Fig. 6 presents the same analysis, but for XLM-R, with PROBELESS ranking (equivalent to Fig. 1 for M-BERT). We see far more overlaps in XLM-R, and no red squares (describing a lower overlap size than the expected one), implying that the information is more condensed in XLM-R than in M-BERT.

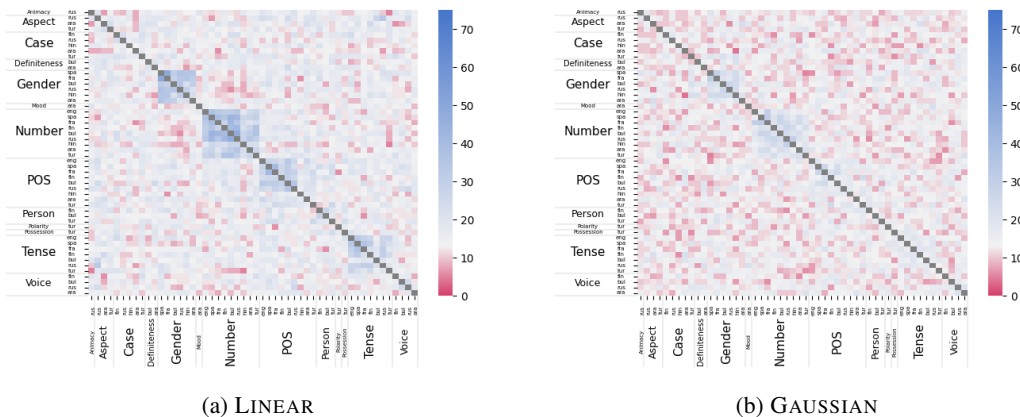

(a) LINEAR          (b) GAUSSIAN

Figure 5: Layer 7 neurons overlap using LINEAR and GAUSSIAN rankings.

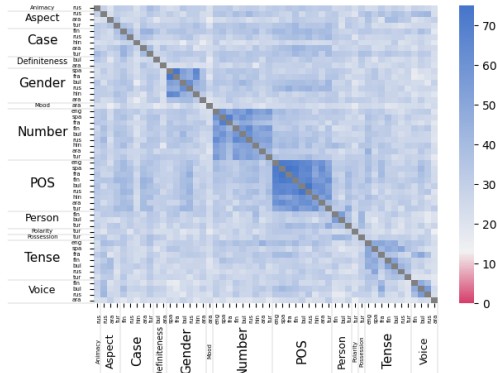

Figure 6: XLM-R layer 7 neurons overlap, using PROBELESS. Blue squares are above expected value, red are below.

## A.5 RANKING EVALUATION BY PROBING

Fig. 7 illustrates the process of evaluating rankings by probing. Two different actors affect the final accuracy: the ranking method, and the probe. In this work, we experiment with the 7 rankings described in §3.1 as ranking methods, and the two probes used by LINEAR and GAUSSIAN as probes. This methodology is problematic for evaluating rankings, as the choice of a probe significantly affects results.

## A.6 CLUSTERING PROBING RESULTS

For each config out of the 156 we experimented with, we have results of 14 classifier–ranking combinations, each of length 150, the max $k$ (number of neurons) we used. For clustering these results, we first remove all combinations involving a bottom-to-top ranking, as these add a lot of noise to the clustering algorithm, making it focus on irrelevant signals. Thus, our results matrix is of shape $[156, 8, 150]$. We then reshape the matrix to shape $[156, 8 \times 150]$ and run K-means over it with $K = 3$. Projecting the K-means output with t-SNE gives us Figs. 3d and 9a.

## A.7 STATISTICAL SIGNIFICANCE TESTS

Table 2 shows the results of our statistical significance tests. The three rows in each cell correspond to using 10, 50 and 150 neurons. If there is an * in the $[i, j]$ cell, is means that the $p$-value under the null hypothesis that probe $j$ is better than probe $i$ is lower than $0.05$, when using the matching number of neurons. For example, we see that there is an * in the first and second rows in the $[0, 3]$ cell,

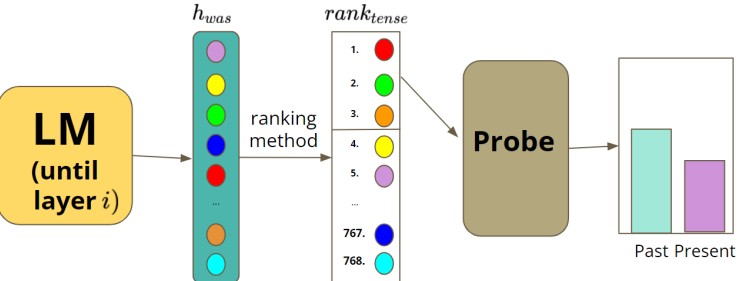

Figure 7: Ranking evaluation by probing: The language model creates a word representation (e.g., of the word "was"), which is fed into a neuron-ranking method, to rank its neurons according to their importance for some attribute (e.g., tense). The $k$-highest ranked neurons are fed into a probe, which is trained to predict the attribute.

Table 2: Statistical significance results. G, L, P and ttb are abbreviations for GAUSSIAN, LINEAR, PROBELESS and top-to-bottom, respectively.

| | G by ttb G | L by ttb G | G by ttb L | L by ttb L | G by ttb P | L by ttb P |
|---|---|---|---|---|---|---|
| G by ttb G | — | * * * | * * * | * * | * * * | * * |
| L by ttb G | | — | * | * | * * | * * |
| G by ttb L | | * | — | * | * * | * |
| L by ttb L | * | * * | * | — | * * | * * |
| G by ttb P | | | * | * | — | * |
| L by ttb P | | * | | * | * | — |

meaning we can confidently reject the hypothesis that LINEAR by LINEAR is better than GAUSSIAN by GAUSSIAN when using 10 or 50 neurons, but we cannot do so for 150 neurons. In fact, looking at the $[3, 0]$ cell shows us that when using 150 neurons, GAUSSIAN by GAUSSIAN is not better than LINEAR by LINEAR.

While we do not show random and bottom-to-top rankings in Table 2 for clarity, we asserted that each classifier is statistically significantly better when using a top-to-bottom ranking compared to a random ranking, and when using a random ranking compared to a bottom-to-top ranking.

### A.8  PROBING: ADDITIONAL RESULTS

Fig. 8 complements Fig. 3, including graph lines that are missing in Fig. 3 due to its readability.

Fig. 9 shows XLM-R probing results. XLM-R provides very similar results to M-BERT, apparent in Figs. 9a and 9b compared to Figs. 3d and 3c, respectively.

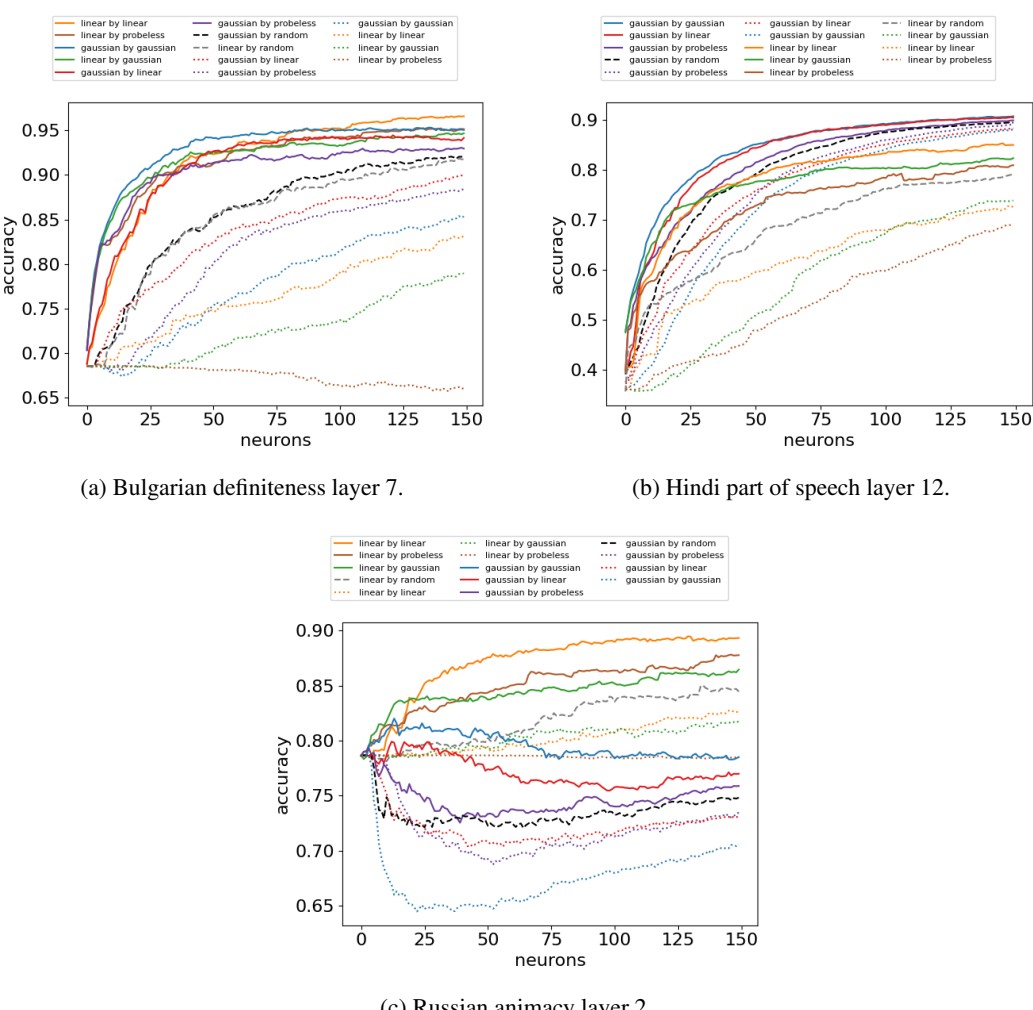

(a) Bulgarian definiteness layer 7.

(b) Hindi part of speech layer 12.

(c) Russian animacy layer 2.

Figure 8: Examples of each of the patterns, with all graph lines (complementing Fig 3). Solid lines are top-to-bottom rankings; dashed are random rankings; dotted are bottom-to-top rankings. "X by Y" means classifier X using ranking Y.

Selectivity examples from both models are provided in Fig. 10. In all configs, both in M-BERT and XLM-R, LINEAR is significantly more selective than GAUSSIAN using any ranking.

### A.9   RANKING EVALUATION BY INTERVENTIONS

Fig. 11 illustrates the process of evaluating rankings by probing. In this methodology, no external probes are involved, and the rankings are evaluated with respect to the degree of their selected neurons importance for the language model's output. In this work, we experiment with the 7 rankings described in §3.1 as ranking methods, we intervene by ablations and translations (§4.1), and evaluate the model's outputs by the error rate and CLWV metrics (§4.3).

### A.10   ABLATION RESULTS

One ablation example is shown in Fig. 12. No matter the ranking, ∼ 400 neurons can be ablated with little impact on the output, and CLWV remains low. This behaviour is generally consistent across all configs we experimented with.

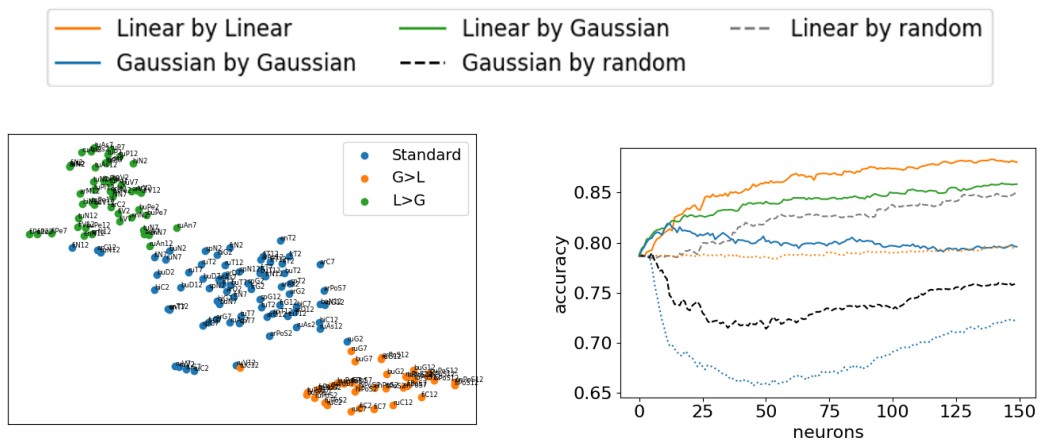

(a) t-SNE projection of clustered probing results, XLM-R.

(b) Russian animacy layer 2 accuracy, XLM-R.

Figure 9: Clustering of the three different patterns in XLM-R, and an example from one config. Solid lines are top-to-bottom rankings; dashed are random rankings; dotted are bottom-to-top rankings. Some lines are omitted for clarity.

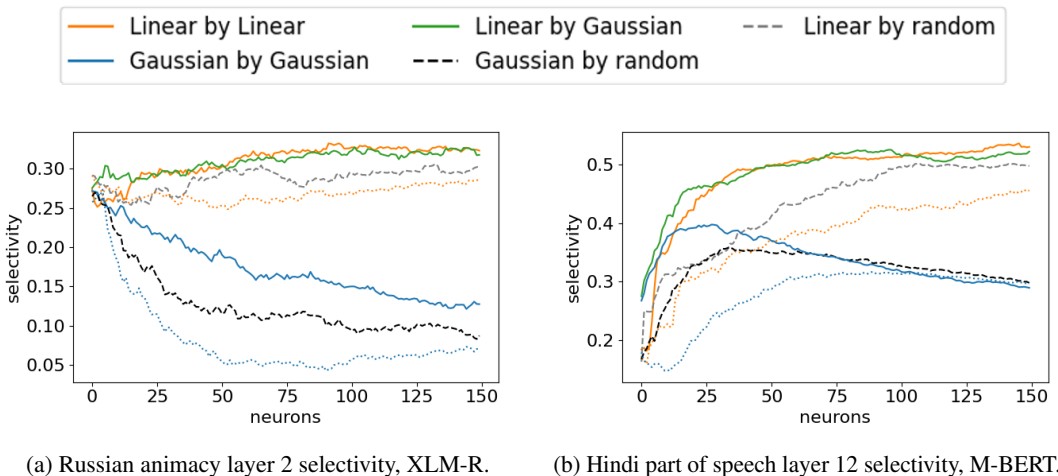

(a) Russian animacy layer 2 selectivity, XLM-R.

(b) Hindi part of speech layer 12 selectivity, M-BERT.

Figure 10: Two selectivity examples from M-BERT and XLM-R. Solid lines are top-to-bottom rankings; dashed are random rankings; dotted are bottom-to-top rankings. Some lines are omitted for clarity.

## A.11 TRANSLATION RESULTS

All of M-BERT's translation results (complementing Table 1) are found in Table 3, and examples from two configs are in Fig. 13. As described in §4.4.3, across most configs, PROBELESS achieves higher CLWV at the saturation point, and gets there earlier (using less neurons), than the other two rankings—in contrast to probing results. Among the probing-based rankings, LINEAR generally provides better results than GAUSSIAN.

We also note that there are certain attributes that seem harder to control for, e.g., English number and French tense.

Table 3: CLWV value at saturation point and number of neurons modified at the saturation point, using the translation method on different configs, with $\beta = 8$. In each cell, the three lines refer to layers 2, 7 and 12 respectively.

| | LINEAR | GAUSSIAN | PROBELESS |
|---|---|---|---|
| English number | $0.04, 70$
$0.09, 50$
$0.11, 130$ | $0.02, 60$
$0.07, 30$
$0.04, 60$ | $0.06, 90$
$0.11, 50$
$0.17, 110$ |
| English tense | $0.39, 60$
$0.37, 50$
$0.51, 60$ | $0.26, 150$
$0.34, 70$
$0.41, 120$ | $0.38, 30$
$0.34, 30$
$0.46, 30$ |
| Spanish number | $0.28, 110$
$0.26, 50$
$0.23, 150$ | $0.19, 100$
$0.20, 40$
$0.16, 140$ | $0.35, 60$
$0.25, 30$
$0.40, 80$ |
| Spanish tense | $0.20, 110$
$0.16, 80$
$0.31, 130$ | $0.15, 140$
$0.11, 70$
$0.18, 70$ | $0.27, 60$
$0.20, 60$
$0.33, 60$ |
| Spanish gender | $0.29, 50$
$0.29, 50$
$0.26, 130$ | $0.25, 80$
$0.31, 50$
$0.16, 110$ | $0.37, 50$
$0.33, 30$
$0.35, 60$ |
| French number | $0.19, 110$
$0.18, 50$
$0.07, 110$ | $0.09, 150$
$0.17, 30$
$0.11, 150$ | $0.25, 60$
$0.20, 30$
$0.33, 120$ |
| French tense | $0.10, 110$
$0.10, 120$
$0.14, 150$ | $0.01, 90$
$0.06, 100$
$0.07, 110$ | $0.13, 70$
$0.08, 70$
$0.15, 90$ |
| French gender | $0.17, 80$
$0.16, 40$
$0.14, 170$ | $0.17, 80$
$0.16, 40$
$0.06, 140$ | $0.22, 60$
$0.17, 30$
$0.20, 60$ |

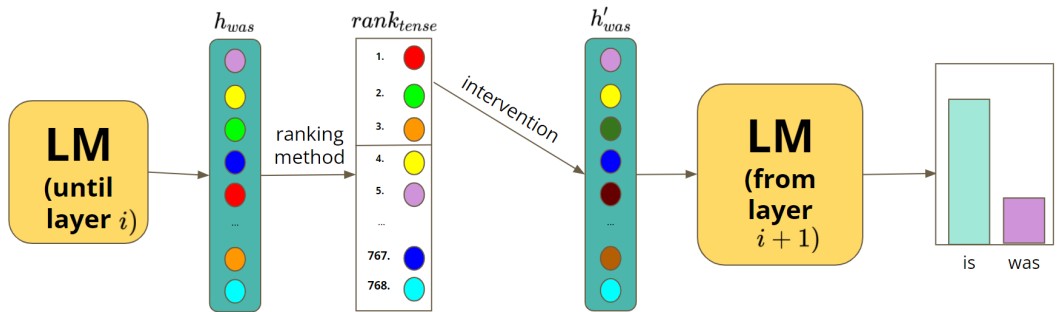

Figure 11: Ranking evaluation by interventions: The language model creates a word representation (e.g., of the word "was"), which is fed into a neuron-ranking method, to rank its neurons according to their importance for some attribute (e.g., tense). The $k$-highest ranked neurons are modified by an intervention (to a different color in the figure), and the new representation is fed into the rest of the language model's layers, to observe the final model's output.

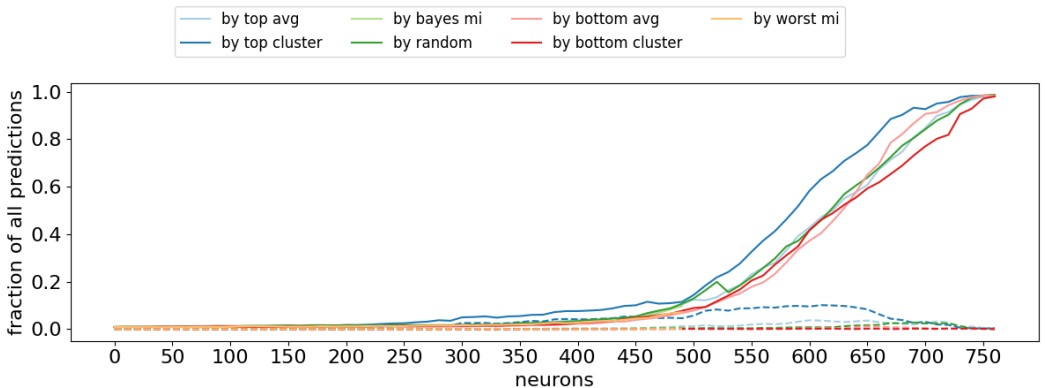

Figure 12: Spanish gender layer 2, ablation results. Solid lines are error rates, dashed are CLWVs.

## A.12 XLM-R TRANSLATION RESULTS

XLM-R translation results (equivalent to Table 3 in M-BERT) are shown in Table 4. The superiority of PROBELESS is not so clear in XLM-R compared to M-BERT, with LINEAR providing good competition. GAUSSIAN on the other hand, still falls behind.

We note that in XLM-R the CLWV values are somewhat lower compared to M-BERT. A possible explanation to that could be the difference in tokenization between the models.

## A.13 TRANSLATION ON MONOLINGUAL MODELS

As LINEAR and PROBELESS are somewhat equal on XLM-R, we perform experiments on additional models, to break the tie. We experiment with three monolingual models: bert-base-cased for English, dccuchile/bert-base-spanish-wwm-cased for Spanish, and camembert-base for French. The results are reported in Table 5. PROBELESS is superior in most of these experiments, both in terms of CLWV value and at number of modified neurons when reaching the saturation point, with LINEAR coming second.

It is worth noting that in these models, saturation point values are generally higher, and achieved using fewer neurons, than in M-BERT and XLM-R. We believe it is due to their relative simplicity compared to multilingual models, and their smaller vocabulary.

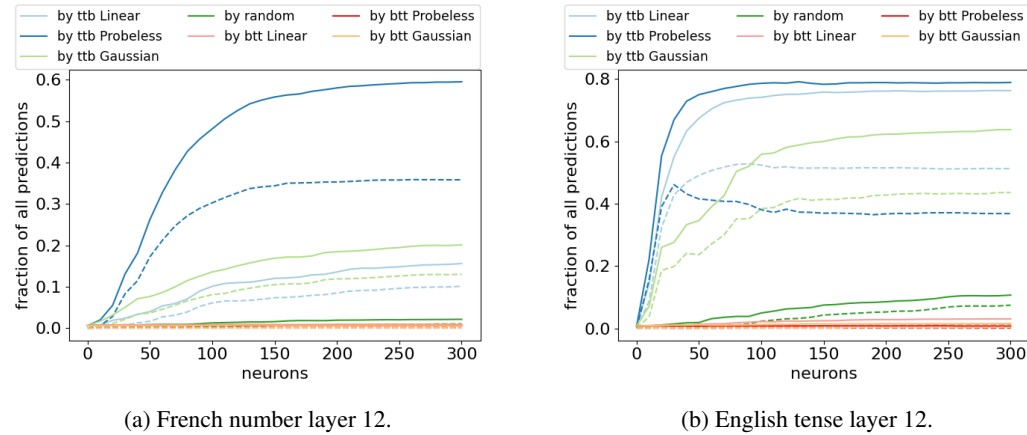

(a) French number layer 12.  (b) English tense layer 12.

Figure 13: Translation results with $\beta = 8$, from two different configs. Solid lines are error rates, dashed are CLWVs. ttb and btt stand for top-to-bottom and bottom-to-top, respectively.

Table 4: XLM-R CLWV value at saturation point and number of neurons modified at the saturation point, using the translation method on different configs with $\beta = 8$. In each cell, the three lines refer to layers 2, 7 and 12 respectively.

|  | LINEAR | GAUSSIAN | PROBELESS |
|---|---|---|---|
| English number | $0.01, 130$
$0.03, 40$
$0.08, 90$ | $0.00, 90$
$0.02, 30$
$0.06, 90$ | $0.02, 80$
$0.04, 90$
$0.08, 90$ |
| English tense | $0.22, 60$
$0.34, 50$
$0.35, 50$ | $0.09, 190$
$0.05, 50$
$0.15, 130$ | $0.21, 30$
$0.09, 60$
$0.19, 50$ |
| Spanish number | $0.29, 70$
$0.23, 30$
$0.33, 60$ | $0.11, 80$
$0.20, 50$
$0.18, 120$ | $0.34, 70$
$0.23, 20$
$0.33, 40$ |
| Spanish tense | $0.08, 90$
$0.22, 50$
$0.16, 60$ | $0.00, 0$
$0.07, 70$
$0.06, 150$ | $0.18, 70$
$0.19, 40$
$0.24, 80$ |
| Spanish gender | $0.36, 60$
$0.39, 70$
$0.39, 60$ | $0.17, 70$
$0.30, 30$
$0.30, 140$ | $0.36, 20$
$0.33, 30$
$0.36, 20$ |
| French number | $0.14, 110$
$0.20, 80$
$0.30, 120$ | $0.06, 100$
$0.14, 60$
$0.11, 130$ | $0.24, 50$
$0.27, 80$
$0.31, 70$ |
| French tense | $0.03, 120$
$0.10, 70$
$0.10, 120$ | $0.01, 120$
$0.01, 40$
$0.02, 110$ | $0.07, 30$
$0.06, 20$
$0.06, 30$ |
| French gender | $0.09, 130$
$0.16, 80$
$0.13, 90$ | $0.02, 150$
$0.06, 70$
$0.05, 100$ | $0.18, 50$
$0.16, 30$
$0.19, 70$ |

Table 5: Monolingual models CLWV value at saturation point and number of neurons modified at the saturation point, using the translation method on different configs with $\beta = 8$. In each cell, the three lines refer to layers 2, 7 and 12 respectively.

|  | LINEAR | GAUSSIAN | PROBELESS |
|---|---|---|---|
| English number | $0.09, 80$
$0.14, 90$
$0.17, 110$ | $0.07, 80$
$0.15, 90$
$0.08, 60$ | $0.09, 40$
$0.17, 60$
$0.22, 90$ |
| English tense | $0.49, 60$
$0.55, 70$
$0.59, 40$ | $0.44, 70$
$0.43, 40$
$0.55, 70$ | $0.47, 20$
$0.50, 20$
$0.55, 30$ |
| Spanish number | $0.57, 30$
$0.44, 20$
$0.49, 80$ | $0.57, 30$
$0.40, 20$
$0.40, 110$ | $0.61, 30$
$0.45, 20$
$0.54, 60$ |
| Spanish tense | $0.41, 60$
$0.41, 90$
$0.47, 90$ | $0.35, 100$
$0.28, 80$
$0.31, 110$ | $0.46, 40$
$0.40, 60$
$0.57, 60$ |
| Spanish gender | $0.38, 30$
$0.35, 30$
$0.34, 110$ | $0.35, 30$
$0.35, 30$
$0.29, 100$ | $0.41, 30$
$0.37, 30$
$0.40, 40$ |
| French number | $0.49, 10$
$0.46, 10$
$0.30, 10$ | $0.50, 10$
$0.50, 10$
$0.31, 10$ | $0.51, 10$
$0.46, 10$
$0.34, 10$ |
| French tense | $0.36, 30$
$0.17, 10$
$0.31, 50$ | $0.20, 50$
$0.26, 10$
$0.13, 100$ | $0.13, 10$
$0.09, 10$
$0.13, 50$ |
| French gender | $0.28, 10$
$0.26, 10$
$0.17, 90$ | $0.28, 20$
$0.26, 10$
$0.14, 20$ | $0.29, 10$
$0.26, 10$
$0.15, 40$ |

