# OpenReview forum: "On the Pitfalls of Analyzing Individual Neurons in Language Models"
_ICLR.cc/2022/Conference — ICLR 2022 Poster_

### Official Review · Reviewer_LRS7 · 2021-11-01

**Correctness:** 3
**Technical Novelty And Significance:** 3
**Empirical Novelty And Significance:** 3
**Recommendation:** 8
**Confidence:** 4

**Main Review:**

The paper begins by first explaining the background of the two methods in the literature, Linear (Dalvi et al 2019) and Gaussian (Torroba Hennigen et al 2020).
Then, the authors describe their proposed "Probeless" method, and then dive deep into the results and how the above two methods suffer in view of the pitfalls.

## The good

One great thing about this paper is how methodical the authors are in their experimental setup. I liked the idea of disentangling the rankings from the classifiers and using them together to see whether an expected ranking (top-to-bottom) is actually better than a worse ranking (random or bottom-to-top). The choice of the metrics also make sense to me. The intervention experiments (Pitfall II) are really interesting, as I think the proposed method could be useful for other related works too. It is indeed very crucial to explore whether the "top-ranked" neurons are actually contributing to the language model output. The results are very interesting - it seems ablation is not a good intervention approach, whereas Translation is a much better strategy. Similar methods have been used in representational probing literature (Amnesic Probing, https://arxiv.org/abs/2006.00995), it would be great to add a note on how they compare.

## The bad

The paper can perhaps present the results/discussion in a better way, as several aspects are not clear to me.

### Section 2.1

- The authors mention they do not exactly use the method shown by Dalvi et al 2019 for Linear, and propose an alternate way to use Linear by computing the mean absolute value of the weights associated with it. Does this enhancement mean it is different from the original Linear model? It would be useful to extrapolate this difference in the paper.
- In the definition of "Probeless", it is not clear what is \hat{z}. Perhaps the authors indicate it is the mean of the words that does not possess the attribute/label information? Maybe I'm missing something obvious.

### Section 2.3

- "We expect to see overlap in the selected neurons" - it is not clear why we should expect to see an overlap among multiple languages. Any explanation/citation to back this claim up?

### Section 3.2

- The results section is a bit hard to parse given the introduction of two new concepts, G>L and L>G. Perhaps the authors should clearly state what these are earlier (it becomes slightly clear after reading Section 3.2.2 that G>L corresponds to Gaussian being better than Linear when it is paired with worse rankings than optimal, and vice-versa))
- Possibly the ordering of the results is incorrect - the t-SNE plot explanation could come after explaining 3.2.1 and 3.2.2

### Section 3.2.2

- Gaussian memorizing in parts-of-speech attribute is expected, as many POS tags can be simply predicted by learning the type of words (POS tagging can be solved by non-neural CRF taggers, and neural probes always gets very high scores in this task anyway). I don't know how this makes Gaussian a "bad" probe. In fact, the inability of learning the basic POS tag associations by Linear probe suggests its weakness, not strength. This suggests a capacity issue.
- Thus, G>L and L>G scenarios are highly task dependent. This calls for their study relative to the task and not in aggregate.

### Section 3.2.3

- The authors say "Probeless provides decent performance (and is inherently consistent)" - however I could not find any results of Probeless probe in Figure 3.

### Section 4.4.3

- I am skeptical of the claims in this section as it seems the numbers are cherry-picked. From Table 1, the Linear probe seems to be better than Probless in English tense. In Appendix Table 3, however, authors show more scenarios where Probeless tops other methods. This table should be moved to the main page.
- However, the claims do not hold well in terms of XLM-R model, as Linear is comparable to the Probless method (gets 3 tasks vs 4 in the latter).

### General comments

- The paper lacks a thorough comparison with recent work. For example, I totally agree in the conclusion that high probing accuracy does not necessarily entail the information is actually useful for the model. Similar investigations have been made in the representational probing literature (Pareto Probing: Trading-Off Accuracy and Complexity, EMNLP 2020)

### Experiment Suggestion

- The authors investigated the results primarily with M-BERT, and they also investigate with XLM-R but the results are not so clear in the latter model. The authors can perhaps also investigate one more similar model, say M2M, to be a tie-breaker for a better conclusion.


**Summary Of The Paper:**

The paper revisits two methods, Linear and Gaussian, as described in the literature, to probe language models based on individual neurons. The paper suggests that these methods contain two limitations - 1. by ranking the neurons on a linguistic task, the methods conflate between ranking quality and the probe's classification quality; 2. the probing methods do not take into account whether the individual neurons are at all used by the model in the downstream linguistic tasks. Finally, the paper presents a new probing method, which does not rely on training a classifier, and shows that this simple method is able to discern among the neurons better than the preceding two methods in the literature.


**Summary Of The Review:**

Overall I found the paper to be quite illuminating on highlighting the different aspects of probing with individual neurons. However, the paper requires more polish in presenting the results and discussions of their work. I believe the finding is important for the community, especially the intervention analysis. However, I'm a bit skeptical of the claims (as highlighted in my review). Improving the writing of the paper and performing an additional model experiment could be useful to make the paper strong for acceptance.

Edit: Following the rebuttal, I'm increasing my original score of 5 to 8

---

> ### Author Response · Authors · 2021-11-17
> **Response to Reviewer LRS7 (1)**
>
> Thank you for your detailed review! We’re glad you found our experiments methodical and liked the idea of disentangling the rankings from the classifiers. We’re happy you found our intervention experiments and their results to be very interesting, and we agree that the proposed method could be useful for other related works too. We're glad you found our paper to be illuminating and important for the community.
> We would like to address your concerns:
> > The authors mention they do not exactly use the method shown by Dalvi et al 2019 for Linear, and propose an alternate way to use Linear by computing the mean absolute value of the weights associated with it. Does this enhancement mean it is different from the original Linear model? It would be useful to extrapolate this difference in the paper.
>
> Thank you for the comment. In our work, we modified the original algorithm slightly, but critically, we use the same probe to generate the ranking. We rephrased this paragraph and provide further details on that in an appendix.
>
> > In the definition of "Probeless", it is not clear what is \hat{z}. Perhaps the authors indicate it is the mean of the words that does not possess the attribute/label information? Maybe I'm missing something obvious.
>
> In our paper there is no \hat{z}, so you probably referred to something else. If you meant z’, it is simply another task label in Z (as shown in Equation 1: z,z’ \in Z).
>
> > "We expect to see overlap in the selected neurons" - it is not clear why we should expect to see an overlap among multiple languages. Any explanation/citation to back this claim up?
>
> Since we experiment with multilingual models, we hypothesize that the model captures a global comprehension of a morphological attribute, meaning it encodes and uses an attribute in a similar way across languages. Consistency across languages suggests that the ranking selects neurons that encode the attribute regardless of the language, which is what we expect to see.
> As far as we know, the evidence for overlaps across languages is a novel discovery, so we don’t know of any citation that supports this claim directly, but consistency across languages is desirable in multilingual models, since those are often used in a zero-shot scenario, fine-tuning in one language and evaluating on other languages (e.g., “Beto, Bentz, Becas: The Surprising Cross-Lingual Transfer Effectiveness of BERT”, ACL 2019 and “COMET: A Neural Framework for MT Evaluation”, EMNLP 2020).
>
> > The results section is a bit hard to parse given the introduction of two new concepts, G>L and L>G. Perhaps the authors should clearly state what these are earlier (it becomes slightly clear after reading Section 3.2.2 that G>L corresponds to Gaussian being better than Linear when it is paired with worse rankings than optimal, and vice-versa))
>
> Thank you for the suggestion. We restructured the opening paragraph of the probing results section, leading with a presentation of the patterns before describing the clustering process. We hope it is clearer now.
>
> > Gaussian memorizing in parts-of-speech attribute is expected, as many POS tags can be simply predicted by learning the type of words (POS tagging can be solved by non-neural CRF taggers, and neural probes always gets very high scores in this task anyway). I don't know how this makes Gaussian a "bad" probe. In fact, the inability of learning the basic POS tag associations by Linear probe suggests its weakness, not strength. This suggests a capacity issue.
>
> Memorization is not necessarily a bad thing. However, the Gaussian method relies on the task performance of the classifier to rank neurons. If the classifier uses memorization for solving the task, it would first select the neurons that are most informative towards the word **identity**, rather than the attribute in question. Also, since memorization is an attribute of the classifier and not of the ranking, it shows the conflation between probe quality and ranking quality, as it makes it hard to decide which ranking is better in these configs.
> The intent of this paragraph is to demonstrate that probe qualities affect ranking evaluation in an undesired manner, and thus this evaluation approach is erroneous. We rephrased this paragraph in hope that the message is clearer now.
>
> > Thus, G>L and L>G scenarios are highly task dependent. This calls for their study relative to the task and not in aggregate.
>
> Aside from POS being in the G>L, we didn’t reveal any other attributes consistently belonging to specific patterns.

---

> ### Author Response · Authors · 2021-11-17
> **Response to Reviewer LRS7 (2)**
>
> > The authors say "Probeless provides decent performance (and is inherently consistent)" - however I could not find any results of Probeless probe in Figure 3.
>
> Since we have 14 probe-ranking combinations, we removed some lines from the graphs as they became hard to read, and thus Probeless is not present in these graphs. However, our statistical significance tests (Appendix A.5) include results from Probeless.
> To ease you concern, to each figure in 3a-3c we uploaded a version that includes all the graph lines to anonymized links.
> Fig. 3a: https://ibb.co/FxrmKLc
> Fig. 3b: https://ibb.co/gZjmWKx
> Fig. 3c: https://ibb.co/1q3Jwzj
>
> > I am skeptical of the claims in this section as it seems the numbers are cherry-picked. From Table 1, the Linear probe seems to be better than Probless in English tense. In Appendix Table 3, however, authors show more scenarios where Probeless tops other methods. This table should be moved to the main page.
>
> For lack of space, Table 1 gives the configs with the highest results overall (across rankings) and not all of the configs. As you noted, it gives a config in which Linear is better than Probeless, emphasizing that these configs were not cherry-picked to show Probeless’ superiority. In Table 3 we see that Probeless consistently achieves higher CLWV than Linear in 6 out of 8 configs, with Linear leading only in one config (English tense), and French tense being a kind of a tie. Nevertheless, in all configs---even those where Probeless does not provide higher CLWV value---Probeless gets to the saturation point earlier, i.e., using fewer neurons, implying that the neurons that are more important for the attribute are indeed ranked higher.
>
> > However, the claims do not hold well in terms of XLM-R model, as Linear is comparable to the Probless method (gets 3 tasks vs 4 in the latter).
>
> We agree that in XLM-R the results are less conclusive in terms of the CLWV value at the saturation point, and we mention it in the paper. However, Probeless still tends to get to the saturation point earlier, meaning it needs fewer neurons than Linear.
>
> > The paper lacks a thorough comparison with recent work. For example, I totally agree in the conclusion that high probing accuracy does not necessarily entail the information is actually useful for the model. Similar investigations have been made in the representational probing literature (Pareto Probing: Trading-Off Accuracy and Complexity, EMNLP 2020)
>
> Thank you for this comment. We added comparison to prior work with regards to this conclusion in section 5. We feel the Pareto Probing work may not be a correct citation here, as it focuses on probe complexity and does not check whether the probed information is used by the model. Aside from that, we do discuss prior work in the introduction, section 4 and in the discussion. We would welcome other suggestions for relevant work.

---

> ### Author Response · Authors · 2021-11-17
> **Response to Reviewer LRS7 (3)**
>
> > The authors investigated the results primarily with M-BERT, and they also investigate with XLM-R but the results are not so clear in the latter model. The authors can perhaps also investigate one more similar model, say M2M, to be a tie-breaker for a better conclusion.
>
> In this work we focus on masked LMs, which have been studied from the perspective of individual neurons in previous work. seq2seq models such as M2M require a different setup and therefore we might experiment with such models in a future work.
> However, we did conduct another set of experiments per your request. Since the only multilingual models supported by huggingface are based on mBERT or XLM, we experimented with monolingual models instead: bert-base-cased for English, dccuchile/bert-base-spanish-wwm-cased for Spanish, and camembert-base for French.
> The results are detailed below. Probeless achieves the highest CLWV value in the saturation in 13 experiments, while Linear does so in 7 and Gaussian only in 2. Also, Probeless gets to the saturation point earlier than the other two in 11 experiments, while Gaussian does so in 2 and Linear never manages to do so.
>
> |       | ttb Linear | ttb Probeless | ttb Gaussian
> | ----------- | ----------- | ----------- | ----------- |
> | English number 2 | 0.09, 80 | 0.09, 40 | 0.07, 80 |
> | English number 7 | 0.14, 90 | 0.17, 60 | 0.15, 90 |
> | English number 12 | 0.17, 110 | 0.22, 90 | 0.08, 60 |
> | English tense 2  | 0.49, 60 | 0.47, 20 | 0.44, 70 |
> | English tense 7  | 0.55, 70 | 0.50, 20 | 0.43, 40 |
> | English tense 12  | 0.59, 40 | 0.55, 30 | 0.55, 70 |
> | Spanish gender 2  | 0.38, 30 | 0.41, 30 | 0.35, 30 |
> | Spanish gender 7  | 0.35, 30 | 0.37, 30 | 0.35, 30 |
> | Spanish gender 12  | 0.34, 110 | 0.40, 40 | 0.29, 100 |
> | Spanish number 2  | 0.57, 30 | 0.61, 30 | 0.57, 30 |
> | Spanish number 7  | 0.44, 20 | 0.45, 20 | 0.40, 20 |
> | Spanish number 12  | 0.49, 80 | 0.54, 60 | 0.40, 110 |
> | Spanish tense 2  | 0.41, 60 | 0.46, 40 | 0.35, 100 |
> | Spanish tense 7  | 0.41, 90 | 0.40, 60 | 0.28, 80 |
> | Spanish tense 12  | 0.47, 90 | 0.57, 60 | 0.31, 110 |
> | French gender 2  | 0.28, 10 | 0.29, 10 | 0.28, 20 |
> | French gender 7  | 0.26, 10 | 0.26, 10 | 0.26, 10 |
> | French gender 12  | 0.17, 90 | 0.15, 40 | 0.14, 20 |
> | French number 2  | 0.49, 10 | 0.51, 10 | 0.50, 10 |
> | French number 7  | 0.46, 10 | 0.46, 10 | 0.50, 10 |
> | French number 12  | 0.30, 10 | 0.34, 10 | 0.31, 10 |
> | French tense 2  | 0.36, 30 | 0.13, 10 | 0.20, 50 |
> | French tense 7  | 0.17, 10 | 0.09, 10 | 0.26, 10 |
> | French tense 12  | 0.31, 50 | 0.13, 50 | 0.13, 100 |
>
> Please let us know if you have any more concerns.

---

> > ### Comment · Reviewer_LRS7 · 2021-11-23
> > **Thank you for the detailed responses and running extra experiments!**
> >
> > As there are multiple replies I'll just summarize my thoughts in one:
> >
> > - Re: Experiments -> Thanks for running these experiments! It would be great to include these in the final draft, probably in the Appendix if short on space.
> > - Re: " If you meant z’," -> Yes this is what I mean't. Its still not clear where this other task is coming from reading the text, please add a one liner explanation in Section 2.1
> > - Re: Explanations on multi-lingual models and memorization -> Thanks for the thorough explanation and restructuring in the paper.
> > - Re: Omitting graphs for Probeless -> While I agree with your logic of removing the Probeless lines to reduce clutter, those are useful (and important) for the reader to understand and analyze the differences in the probes. It is indeed hard to parse (from your new figs), so probably it would be beneficial to use the extra space to incorporate the Probeless performance somehow.
> > - Re: Table 1 -> My point on cherry picking was that you can easily just replace Table 1 with the full table (Table 3) as it provides a more comprehensive view (and fair view) of the performance of Probeless compared to the rest. You could also bold the numbers across each config (both in CLWV value and saturation point neurons) to show them more clearly.
> >
> > Overall, I'm happy with the rebuttal as it explains my questions. I'm also glad you added another model experiment in this short time, adding those results in the paper would be helpful for future readers. Aside from the minor points mentioned above, I don't see any more major problematic issues with the paper or presentation. I'm happy to improve my score to 8.

---

> > > ### Author Response · Authors · 2021-11-23
> > > **Thank You**
> > >
> > > Thank you for your comment and for improving your score! We're happy you found our extra experiments helpful, and agree these should be added in the final draft. We will incorporate this and your other remaining comments in the final version of the paper.

---

### Official Review · Reviewer_RqKU · 2021-11-02

**Correctness:** 3
**Technical Novelty And Significance:** 3
**Empirical Novelty And Significance:** 3
**Recommendation:** 8
**Confidence:** 3

**Main Review:**


Strengths:
- PROBELESS seems to be genuinely an extremely good neuron ranking method and is simple and intuitive.
-  The new methods for evaluating neuron ranking could be very useful, especially if the experimenter is interested in rankings that specialize in a particular task.
- I was very interested in the finding that gaussian models memorize the data, likely because they rely on a smaller number of potential features.
-  The analysis is extremely in depth. It provides both conjectures and well justified explanations for why certain rankings outperform others.
-  I particularly like the analysis of the translation method of removing neurons, because it uses the saturation point where the lemma starts to disappear, rather than a single arbitrary point or some kind of AUC.
-  The paper was generally clear to me.

Weaknesses:
- It’s not clear to me that the linear and gaussian methods specifically aim to find neurons that specialize (instead possibly targeting much less specialized but significant neurons), whereas their probeless method clearly targets neurons specializing in a particular task label. However, their metric is based on the neuron’s significance for a particular label, while controlling for preserving lemma information, which is not clearly the goal of the other methods. The major problem in this paper is that the decisions made in evaluation and ranking are not entirely well justified.
-  I’m not convinced that it’s fair to evaluate rankings based on whether the rankings assigned by one probe are useful to a different probe.
- I would like to see some discussion of why single neuron analysis actually is preferred over analyzing distributed dimensions. In particular, the fact that translation outperforms the ablation method does indicate that the attributes in question could be encoded in a weighted distribution over neurons rather than directly into single neurons.
- On page five, the authors hypothesize that the linear  probe method is more table because only a few dimensions are informative and gaussian distributed. I would consider this a conjecture, but I would like to see experiments that might support this hypothesis.

Minor:
- I would have liked to see an explanation of the tasks that were used in section 2.2.

QUESTIONS:
- How are A, L (the attribute and lemma labeler) trained?


**Summary Of The Paper:**

This paper responds to several recent works focused on identifying important individual neurons for particular classifying tasks. They consider 2 existing methods which rely on an external probe to rank the neurons in a network. They also introduce a method that does not rely on a probe, instead ranking neurons according to the difference between their values across labels.

The primary focus in this paper is on two claimed flaws of the existing neuron ranking methods. The first is that the authors feel evaluation of the neuron rankings is unfair because a high quality probe can have higher performance on a worse ranking, simply because it is better able to take advantage of the data that is offered. The second is that not all rankings actually point to neurons that are specialized for a particular task or label, instead just indicating highly informative neurons.

In analyzing these two flaws, the authors develop evaluation metrics for the rankings of these neurons. They consider a ranking better if the neurons encode information that is specific to the label while maintaining the particular lemma being encoded. They also consider the ranking to be better if removing the features indicated damages the performance of the language model. They test two ways of modifying the network to remove important neurons, first by ablating the neurons and then by learning a geometric translation that moves a word towards a different attribute label.

They also consider the ranking better if a ranking from top to bottom outperforms a random ranking, which outperforms a reversed ranking. This seems like a fairly weak standard to hold a ranking to, but some rankings fail to adhere to it.

**Summary Of The Review:**

This paper is an in-depth analysis with new perspectives on how we should think about neuron ranking. It does not discuss much the difference between their philosophy towards the purpose of neuron ranking, compared to the implicit philosophy in the prior works which they compare. Overall it seems valuable but may not be entirely fair to the existing literature.

---

> ### Author Response · Authors · 2021-11-17
> **Response to Reviewer RqKU**
>
> Thank you for taking the time to review our paper! We’re glad you found our new ranking method to be “extremely good, simple and intuitive”. We agree that our ranking evaluation method, by interventions, can be very useful to other works in the field. We’re happy to hear you appreciate our in-depth analysis and the choice of saturation point, and thought the paper was clear.
> We would like to address your concerns:
>
> > It’s not clear to me that the linear and gaussian methods specifically aim to find neurons that specialize (instead possibly targeting much less specialized but significant neurons), whereas their probeless method clearly targets neurons specializing in a particular task label. However, their metric is based on the neuron’s significance for a particular label, while controlling for preserving lemma information, which is not clearly the goal of the other methods. The major problem in this paper is that the decisions made in evaluation and ranking are not entirely well justified.
>
> Critically, all three methods were designed for the same task: to rank neurons according to their importance for a **specific attribute**, and not according to their general importance.
> It is correct that prior methods are not designed specifically for identifying neurons that hold attribute information and do not hold other information, and neither is Probeless. However, we feel it is a desired trait of a ranking method: to isolate specific attribute information from other information, since it can be used for causal analysis and a fine-grained control over the model’s outputs. While the methods are not directly designed for that, we feel it is important to compare them in this regard.
>
> > I’m not convinced that it’s fair to evaluate rankings based on whether the rankings assigned by one probe are useful to a different probe.
>
> The main idea of this comparison was to show that the evaluation of the rankings depends on the probe that is used for evaluation, making it a problematic evaluation method, as demonstrated by patterns G>L and L>G. Yet, using this method we do observe an interesting phenomenon: that the Linear classifier provides better accuracy using the Gaussian ranking rather than the Linear ranking for small sets of neurons (3.2.3).
>
> > I would like to see some discussion of why single neuron analysis actually is preferred over analyzing distributed dimensions. In particular, the fact that translation outperforms the ablation method does indicate that the attributes in question could be encoded in a weighted distribution over neurons rather than directly into single neurons.
>
> From the perspective of localization vs. distributivity, both ablation and translation work the same: they both modify the values of exactly $k$ neurons, leaving the other $d-k$ neurons as they are. The only difference between the two is that while ablation sets the values of all these k neurons to $0$, translation gives them different values based on the distribution.
> As for the motivation for analyzing individual neurons rather than distributed dimensions: looking into individual neurons is more intuitive and interpretable, and can be used directly for pruning and controllability. We discuss these aspects in the introduction.
>
> > On page five, the authors hypothesize that the linear probe method is more table because only a few dimensions are informative and gaussian distributed. I would consider this a conjecture, but I would like to see experiments that might support this hypothesis.
>
> This was actually shown as a limitation in the original Gaussian paper (Torroba Hennigen et al., EMNLP 2020, section 6.1 and Figure 6). We rephrased this sentence to reflect that.
>
> > I would have liked to see an explanation of the tasks that were used in section 2.2.
>
> We rephrased to make it clear that the tasks are morphological attribute prediction, and added a list of the attributes we work with in the appendix.
>
> > How are A, L (the attribute and lemma labeler) trained?
>
> As pointed in section 4.2, for both A and L we use the pre-trained models provided by spaCy.
>
> Please let us know if you have any more concerns.

---

> > ### Comment · Reviewer_RqKU · 2021-11-17
> > **Seems good**
> >
> > Thanks for clarifying about translation. I understand the method better.
> >
> > If there's something missing from the paper right now, I think it would be a more explicit description of what you perceive as the purpose of isolated neuron ranking systems, early on in the paper. As is, the purpose seems to unfold mostly from how these methods are measured, meaning that the metrics you use end up being the description for what they are trying to measure. I don't find the metrics unconvincing, but there isn't a clear explanation of what they are trying to measure before the experiments are introduced.

---

> > > ### Author Response · Authors · 2021-11-18
> > > **Thank You**
> > >
> > > Thank you for your comment and suggestion. We rephrased the paragraph describing Pitfall II in the introduction to better reflect the motivation for these experiments.
> > >
> > > Please let us know if you have any more concerns.

---

### Official Review · Reviewer_zE55 · 2021-11-04

**Correctness:** 4
**Technical Novelty And Significance:** 3
**Empirical Novelty And Significance:** 3
**Recommendation:** 8
**Confidence:** 3

**Main Review:**


# Strengths

This paper identifies issues with existing methods for ranking neurons containing linguistic attributes and presents a detailed experiment showing it empirically. To address these issues, a new ranking method is proposed and it is effective.

# Weaknesses

n/a
This paper identifies issues with existing methods for ranking neurons containing linguistic attributes and proposes an effective solution.


**Summary Of The Paper:**

This paper identifies two pitfalls of existing methods for ranking individual neuron contributions to certain linguistic attributes, and shows empirically that these pitfalls indeed exist. Additionally, it proposes a new ranking method free of these pitfalls, and shows its effectiveness.


**Summary Of The Review:**

This paper identifies issues with existing methods for ranking neurons containing linguistic attributes and proposes an effective solution.

---

> ### Author Response · Authors · 2021-11-17
> **Response to Reviewer zE55**
>
> Thank you for taking the time to review our paper. We’re glad you liked it.

---

### Official Review · Reviewer_C78M · 2021-11-06

**Correctness:** 4
**Technical Novelty And Significance:** 3
**Empirical Novelty And Significance:** 3
**Recommendation:** 6
**Confidence:** 3

**Main Review:**

Strengths:
 - The paper describes flaws in multiple recent high-profile NLP interpretability papers
 - The paper proposes an alternative mechanism (PROBELESS) which behaves differently from existing methods
 - Models studied are cross-lingual, unlike most probing work which has focused on English
 - Ranking and intervention experiments are generally convincing and well-executed

Weaknesses:
 - Generally I found this paper was difficult to read. For example, the conflation between probe quality and ranking quality was not clear to me in the abstract/introduction, and perhaps an illustrative example or figure would help clarify this point. Another option would be to clarify >>early on<< that rankings are non-optimal due to e.g., greedy inference procedures which do not search over the whole space of rankings. Similarly, a figure representing the intervention procedure and CLWV metric might be useful.
 - Pitfall II is not especially novel, since it is well-known that most probing methods in NLP are fundamentally correlational in nature. Despite this, findings about the PROBELESS model and interventions are a useful contribution to the literature and fall in line with other work (e.g., ICLR paper from Lovering, et al. 2021) showing that extractability of information correlates with its usefulness in model predictions.

Questions:
 - Why is consistency across languages a desirable attribute here? (End of Section 2.2)
 - Can you re-explain what I'm supposed to be getting out of Figure 3a? Must be missing something.
 - In 3.2.2 it seems that "Gaussian is memorizing" is posed as a negative, but this is not evident to me. Why would memorization in this case (of a small set of function words) be considered a bad thing?

Writing Comments/Nits:
 - In the introduction: "We see this framework as <exhibiting> Pitfall I"
 - The following sentences from Section 2.1 are difficult to interpret without reading the referenced paper: "However, we observed that the original algorithm distributes the neurons equally among labels, meaning that each label would contribute the same number of neurons at each portion of the ranking, regardless of the amount of neurons that are actually important for this label"
 - "PROBELESS prefers neurons that separate different labels over neurons that have a similar value across labels" could be rewritten as something like "PROBELESS assigns high values to neurons which are most sensitive to a given attribute"
 - Appendix A.2: doesn't this assume that all rankings of PROBELESS are equally likely? This might not be true if some neurons vary more than others, for example. [Please correct me if I am wrong.]

**Summary Of The Paper:**

This paper describes two issues with existing probing methods within the NLP interpretability space and proposes an alternative approach which does not exhibit the same flaws.

**Summary Of The Review:**

Although the paper is somewhat difficult to read (see suggestions), it provides a useful new approach to single-neuron probing and summarizes some weaknesses in recent existing work.

---

> ### Author Response · Authors · 2021-11-17
> **Response to Reviewer C78M (1)**
>
> Thank you for the helpful review! We’re glad you found our novel method useful, were convinced by our experiments and appreciated the critical view on recent work and the cross-lingual experiments.
> We would like to address your concerns:
> > Generally I found this paper was difficult to read. For example, the conflation between probe quality and ranking quality was not clear to me
> in the abstract/introduction, and perhaps an illustrative example or figure would help clarify this point. Another option would be to clarify >>early on<< that rankings are non-optimal due to e.g., greedy inference procedures which do not search over the whole space of rankings.
> Similarly, a figure representing the intervention procedure and CLWV metric might be useful.
>
> Thank you for these suggestions. We added a clarification about the non-optimality of ranking methods in the introduction (third paragraph). Due to space limitations, it is complicated for us to add such figures, but we will still look into this possibility until the submission deadline.
>
> > Pitfall II is not especially novel, since it is well-known that most probing methods in NLP are fundamentally correlational in nature. Despite this, findings about the PROBELESS model and interventions are a useful contribution to the literature and fall in line with other work (e.g., ICLR paper from Lovering, et al. 2021) showing that extractability of information correlates with its usefulness in model predictions.
>
> We concur that prior work acknowledged the correlational nature of probing classifiers, but neuron-level work, which we study, has largely not internalized this criticism. Prior work such as the TACL paper “Amnesic Probing” from Elazar et al. 2021 uses the full representation rather than individual neurons. We added a discussion about this conclusion being present in prior work (but from a different perspective) in the discussion section.
> Thank you for the citation suggestion, we added it in section 4 opening.
>
> > Why is consistency across languages a desirable attribute here? (End of Section 2.2)
>
> Since we experiment with multilingual models, we hypothesize that the model captures a global comprehension of a morphological attribute, meaning it encodes and uses an attribute in a similar way across languages. Consistency across languages suggests that the ranking selects neurons that encode the attribute regardless of the language, which is what we expect to see.
> Moreover, consistency across languages is desirable in multilingual models, since those are often used in a zero-shot scenario, fine-tuning in one language and evaluating on other languages (e.g., “Beto, Bentz, Becas: The Surprising Cross-Lingual Transfer Effectiveness of BERT”, ACL 2019 and “COMET: A Neural Framework for MT Evaluation”, EMNLP 2020).
>
> > Can you re-explain what I'm supposed to be getting out of Figure 3a? Must be missing something.
>
> In the revised version this is Figure 3d.
> Each point in Figure 3d corresponds to probing results from one config. For example, one of the blue points in 3d represents the config from 3a, one of the orange points represents the config from 3b, and one of the green points represents the config from 3c. Since we have 156 configs overall, Figure 3d provides a categorization of the configs to the three patterns, and shows that about half of the configs follow the standard pattern, and the G>L and L>G patterns constitute the other half.
>
> > In 3.2.2 it seems that "Gaussian is memorizing" is posed as a negative, but this is not evident to me. Why would memorization in this case (of a small set of function words) be considered a bad thing?
>
> Memorization is not necessarily a bad thing. However, the Gaussian method relies on the task performance of the classifier to rank neurons. If the classifier uses memorization for solving the task, it would first select the neurons that are most informative towards the word identity, rather than the attribute in question.
> The intent of this paragraph is to demonstrate that probe qualities affect ranking evaluation in an undesired manner, and thus this evaluation approach is erroneous. We rephrased this paragraph in hope that the message is clearer now.
>
> > The following sentences from Section 2.1 are difficult to interpret without reading the referenced paper: "However, we observed that the original algorithm distributes the neurons equally among labels, meaning that each label would contribute the same number of neurons at each portion of the ranking, regardless of the amount of neurons that are actually important for this label"
>
> Thank you for the comment. In our work, we modified the original algorithm slightly. We rephrased this paragraph and provide further details on that in an appendix.

---

> ### Author Response · Authors · 2021-11-17
> **Response to Reviewer C78M (2)**
>
> > Appendix A.2: doesn't this assume that all rankings of PROBELESS are equally likely? This might not be true if some neurons vary more than others, for example. [Please correct me if I am wrong.]
>
> In Appendix A.2 we calculate the expected overlap between two random rankings, not necessarily those provided by Probeless. This is done in order to put any overlap result in context (whether there is high or low overlap).
>
> Please let us know if you have any more concerns.

---

> ### Author Response · Authors · 2021-11-21
> **Added Figures**
>
> Hi, we wanted to let you know that we added two figures in an updated version of the paper, illustrating the ranking evaluation by probing process (pitfall I) and ranking evaluation by interventions (our methodology for pitfall II), in an appendix. We will try to fit those in the main body if we have space.
>
> For your convenience, the added figures are also available here:
>
> probing figure: https://ibb.co/BZ0Xntk
>
> interventions figure: https://ibb.co/WG06cR5

---

> ### Author Response · Authors · 2021-11-25
> **Any Other Issues?**
>
> Hi, as the discussion deadline is getting close, we would like to know whether we addressed your concerns, or if there are any more issues we should clarify.
> We would be happy to hear from you.

---

### Decision · Program_Chairs · 2022-01-20

**Decision:**

Accept (Poster)

**Comment:**

This paper analyses interpretation methods that use probes to evaluate the information in individual neurons of a deep network and shows that it confounds probe quality and ranking quality, and encoded information and used information. The paper proposes a new method which does not suffer from the same drawbacks. The reviewers were positive about this paper, and the discussion between the reviewers and authors resulted in the authors adding multiple clarifications. I ask the authors to try to optimize the paper for clarity further. I recommend acceptance.